# Gender, Migration, and Drought: An Exploratory Study of Women's Roles in Mallee Farming Communities

**Anna Kosovac[1], Madeline Grupper [2]**

[1] Faculty of Arts, The University of Melbourne, Australia
[2] Faculty of Engineering and Information Technology, The University of Melbourne, Australia

Correspondence to: Anna Kosovac (anna.kosovac@unimelb.edu.au)

**Abstract.** A gender difference exists in the access to resources and inclusion in decision-making in issues of drought as women are overwhelmingly denied a 'voice' in such a landscape (Clarke 2014). This is particularly prevalent in irrigation and farming communities which carry on a legacy of patriarchal stewardship over farming and agricultural matters. This exploratory study considers the role of women in farming practice in the Mallee Region and how they view their position as decision-makers in drought and water management. This study presents three key findings from interviews of women within the region: women are increasingly adopting the label 'farmer' so that they can be 'counted' and given decision-making power regarding drought and water. Interviewees also stated a distinct difference in gender relations within horticultural dryland farming, compared to irrigation farming. Namely, many found that that gender dynamics were more progressive and equal within dryland. Some stated that this was due to many irrigation farmers being recent migrants and more likely to have traditional gender roles in their own family units, resulting in a perceived subordination of women. The dynamic between white settler farming women and those who had recently settled in the area (first generation migrants) was wholly unexpected and highlights a potential 'us-and-them' distinction in farming. Despite the psychological distance of drought during the time of the interviews (many had recently experienced flooding), there was nevertheless a strong sense of the danger of drought, and the foreboding sense that it was coming. Interviewees stated that women were pivotal during times of drought as they were the ones to draw on community networks for help, to apply for grants, and also to supplement family income from off-farm work. This research should be noted for its limitations, particularly regarding the low sample size. As an exploratory study, it cannot be said to be representative and as such, can only present potential areas for future research.

## 1 Introduction

Women are hitting the 'grass' ceiling in agriculture. As coined terms go, Margaret Alston (Alston, 2013 (2000)) has hit on a pun that both reflects the position of women in farming, while also encompassing the intractability of an issue that extends to all members of the agricultural community. In Australia, farming culture is rooted in the duality of being adaptive to environmental change while staying true to post-colonial social traditions and historical roots (Rodriguez Castro and Pini, 2022; Alston, 2021). These cultural imperatives are challenged on both fronts by environmental change and identity shifts about who gets to be a 'farmer' (Rodriguez Castro and Pini, 2022). For a long time, social tradition elevated white men in

decision making spaces (Rodriguez Castro and Pini, 2022), but there has been recent encroachment as other groups, such as recent migrants and larger corporations, have attempted to take up the farming mantle. At the same time, climate change has

created unprecedented challenges to farmers' ability to maintain their land as the deluge of flooding, interspersed with droughts, results in additional challenges related to agriculture. These juxtaposing effects (overly wet and overly dry) introduce a question of how differing perspectives, particularly related to gender, may impact farming culture and resilience in the face of environmental change.

Settler farming experiences in Australia are socially gendered, resulting in predetermined expectations in roles based on gender

assignation (Twigg, 2021). Settler farming women have almost always been considered in traditional gender roles such as the 'farmer's wife' or the 'haggard woman,' (Twigg, 2021) which places women in positions of disempowerment that facilitates ongoing dominance of men in farming (Whatmore, 1991). These images associate women with exhaustion and subservience, despite women often carrying out administrative or field work essential to survival of the farm (Rickards, 2008; Twigg, 2021; Alston and Whittenbury, 2013). In contrast, there has been an overarching mythologisation of the farmer as a battler, carrying

discourses of survival, persistence, stoicism and struggle (Bryant and Garnham, 2015). This rhetoric often puts settler "hegemonic masculinity" (see Raewyn Connell's (1995) conceptualisation of this) and perspectives of men on a pedestal. Masculine hegemony is recognised as the most pervasive influence on drought rhetoric and discussion of regional water issues in Australia (Clarke, 2014; Holmes, 2017) and has been recognised within academic feminist social critique as omitting and making invisible women's experiences in agricultural communities (Alston, 2006; Rickards, 2008; Alston, 2021).

Despite women in Australia making up 32% of the farming workforce (Alston, 2013 (2000)), they are often denied access to resources and inclusion in decision-making in issues of environmental change, with women overwhelmingly denied a 'voice' in such a landscape (Clarke, 2014; Zwarteveen, 2008). Similarly, women lack representation on boards and within water organisations, limiting their access to decision making spaces that impact policy and management. This also carries to the public domain, with a recent study by Kosovac et al (2024) demonstrating that men have had the most prominent public 'voice'

in irrigation and environmental water debates and were given the widest media platform from which to present their perspectives on water issues. This carries implications for environmental decision-making as women tend to have a more pro-environmental lens when making choices (McCright and Xiao, 2014; Casey and Scott, 2006). In turn, masculine-dominated water management practices have tended to focus on technological solutions to environmental problems. This trend carries implications for centring technocratic solutions that may have limited benefit for both women and the environment more

generally (see Kosovac, 2021 for full argument).

The continuing trend of patriarchal hegemony in settler farming has left little legitimacy for women looking to establish themselves as 'farmers.' Women have accepted less visible workloads, often undertaking unpaid farm work in addition to family upkeep. This lack of visibility, voice, and image has implications for justice for women looking to establish themselves as new farmers. Farmland has also often been kept within the family and inherited through generations, but inheritance of

family farming practices tends to pass over women in favour of family members who are men (Carolan, 2018).

Decreased legitimacy for settler women in farming is not only relevant for equity's sake; it bears implications for environmental trends and adaptation to change. Australia has seen both drought-ridden and flooded landscapes, prompting financial and psychological difficulty for farmers (Heo et al., 2020). With climate change, the risk of future drought is inevitable, but recent La-Ninã years may have driven that risk out of sight and consequentially out-of-mind, resulting in low salience (Stewart, 2009). This study explores the decision-making power farmers who identify as women feel they—and others—have on their farms, particularly in the context of drought. However, as this this is an exploratory study, it does have a low sample size. As such, we suggest this study is taken as an initial step, rather than a representative finding and to create opportunities for further research.

## 2 Study Area

Our work focuses on the Mallee region of southern Australia (see Figure 1). This region had been cared for and occupied for thousands of years by various Traditional Owner groups of the region, including (but not limited to) Latji Latji, Wadi Wadi, Wamba Wamba, Tati Tati, Jari Jari, Nyeri Nyeri, Ngintait, Ngarkat and Barengi Gadjin Land Council Aboriginal Corporation (Mallee Catchment Management Authority, nd). For thousands of generations, these tribes lived off roots, berries, and grass seeds (Gardner, 1986). From the 1850s, and the onset of Europeans in the area, white settlement in the region significantly expanded. A surveyor, exploring the area in 1864 recounted that "I can readily imagine why most people speak of this part of the country with a certain dread for there is actually no grass and no water to be found" (Victorian Historical Journal, 1975).

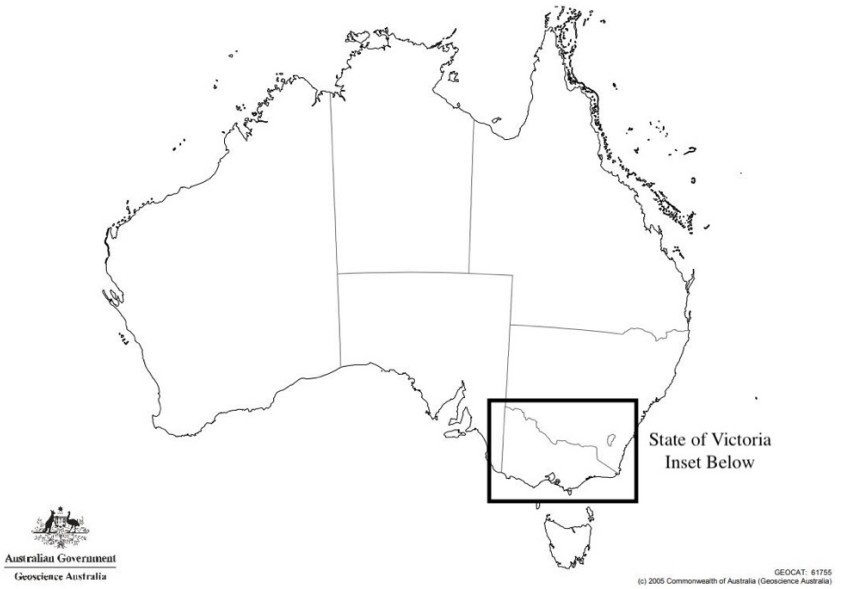

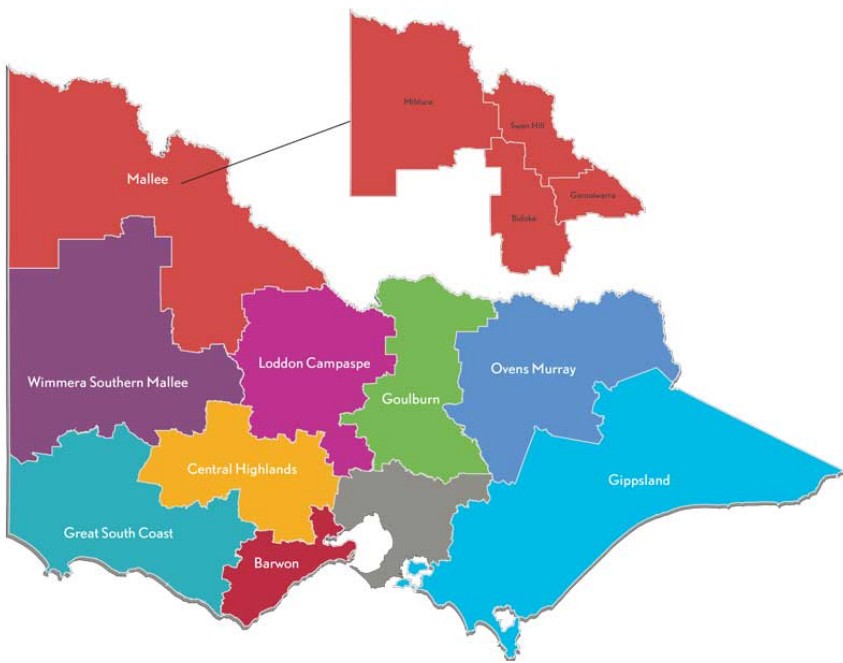

Figure 1. Map of Australia (Source: Australian Government GeoSciences Australia, 2005) and the Mallee Region, Australia. Source: Regional Development Victoria, nd.

Many rural disasters have plagued the region, including the Federation Drought between 1895 to 1903 and the more recent, Millennium drought between 1997 and 2009. A 'drought' is defined by the Australian Bureau of Meteorology (nd) as "prolonged, abnormally dry period when the amount of available water is insufficient to meet our normal use". The onslaught of droughts and dust storms in the region severely limited the capacity of farmers to be able to continue their practices, with graziers walking off their land, and pastoralists overwhelmed by debt. Notable Australian poet, Banjo Paterson, writes in 1902 of the central role that water plays in the drought-stricken region:

"It's grand to be a Western man, with shovel in your hand, to dig your little homestead out, from underneath the sand… It's grand to be a lot of things in this fair southern land, but if the lord would send us rain, that would, indeed be grand!"

In 1887, the Victorian government implemented a large-scale irrigation scheme in the region, sourcing their water from the Murray-Darling Basin river system. Increasing extraction from the river has resulted in a decline in the river water quality (salinity issues) and in the flora and fauna that use the riparian zones for their livelihoods (Kosovac et al., 2023). A market-based water allocation scheme was introduced in the 2012, which separated land rights from water rights, allowing water rights to be bought and sold on the market as needed (see Loch et al (2017) for more information on this). Although drought has always been a feature of the region, in recent years there has been a deluge of flash flooding taking its place instead. Inundation

of towns such as Mildura and Swan Hill had been a feature of the landscape since 2020 and presents the backdrop to the interviews taking place in this study.


Within this work, it is important to remain cognisant of the differences between the farming practices. Irrigated agriculture and dryland farming are the two main types of agricultural practices in the Mallee region. Their main differences exist in the water required for crop production. Irrigated agriculture involves the controlled application of water to crops, typically through artificial means such as sprinklers, drip irrigation, or flood irrigation (Kirby, 2011). This allows crops to be grown in areas
with limited rainfall or in regions where rainfall is unevenly distributed. Dryland farming, in contrast, refers to crop production in areas where rainfall is the primary source of water for plants. This type of farming is more dependent on natural precipitation and is typically practiced in areas where rainfall is sufficient to support crop growth. Farmers often rely on techniques such as crop rotation, soil conservation, and drought-resistant crop varieties to maximize yields and minimize the impact of droughts (Kirby, 2011). Types of farming practice not only affects farmer relationships to water, but also the types of bonds within
respective communities. Practice structures differ substantially, with irrigation being heavily influenced by larger corporations whereas dryland agriculture remains largely in the realm of small business/family farming. Within irrigated agriculture, there is a move to the corporatization of farming over the last decade which has reduced the proportion of 'family-led' farming in this space. This has resulted in gender dynamics to be played out in 'professionalised' settings that carry their own barriers in women's representation in leadership practices (Sheridan and Newsome, 2021).

**3 Methods**

To understand further questions related to the extent of women's perceived 'voice' in drought and farming practice, we conducted semi-structured interviews with those linked to farming who identified as women in the Mallee region (n=6). The sampling for the study relied on existing networks at the Mallee Regional Innovation Centre (MRIC), a partnership between the University of Melbourne, La Trobe University and SuniTAFE. Based in Mildura, this centre works closely with local
growers in the Mallee region to achieve agricultural sustainability across the region. It has strong connections with the local community. We consulted with the MIRC to identify and recruit community members who identified as women, lived in the Mallee region, and had links to agriculture and farming in the region. We conducted interviews online instead of in-person due to researcher constraints on travel, a factor which may have restricted the number of participants. Furthermore, the timings of the interviews were in February/March of 2022 which was harvesting season for many of the growers in the region, once again
limiting participation rates. These exist as limitations to the study that should be considered for any future research in the region.

The small sample size was decided upon early in the project to account for a range of factors: limited funding, timing changes due to COVID lockdowns, caring responsibilities which limited travel, and also the consideration (as reflected by the MRIC) that there is a sense in the community of being 'over-studied'. As such, we decided to pivot this study to be one that is

exploratory, rather than one that will necessarily provide the wide representativeness of larger-scale approaches. There are many studies that reflect a smaller sample size that are nonetheless incredibly valuable for their insights, for example, Young and Casey (2019) examine a range of qualitative research projects to determine at what point they achieved saturation of themes. They found that the majority of projects they surveyed had 100% of the themes covered by n=5 or n=6. This, however, does not presuppose that all projects will necessarily reach saturation point by such a small sample size, but it does still

highlight that these types of studies can still be meaningful, especially in their ability to be explorative and provide opportunities for further study.

Our interview approach was chosen for its conversational style to de-limit the responses of the people being interviewed. The broad questions (Appendix A) were developed as a guide to provide an opportunity for research participants to show their world in a way that is flexible and reflective (Bryman, 2016). These questions were developed from the themes in the literature

around gender and drought, most notably, their feelings of 'having a voice' on climate change, roles within the family, framing risks of drought, and barriers to decision-making/empowerment.

We interviewed six participants, four were directly involved in farming practice through their own or a family farm (two from irrigation farming, two from dryland), one was involved in conservation work, and one was involved in providing financial counselling to farming families stricken by drought. We recorded the interviews online, transcribed them using automatic

transcription software (Trint), and qualitatively coded them in NVivo by way of thematic analysis (Bryman, 2016). The coding was purely inductive, that is, drawn directly from the data. Codes that were similar were merged (see Appendix B for the full list of codes).

After developing the list of the main thematic findings from interview coding, we presented these themes to the MRIC working group to discuss our findings. These discussions were purely engaged as a 'check' to compare their extensive local knowledge

to our interview findings and interpretations. The working group comprised of representatives from peak industry bodies, horticultural organisations, farmers and local business owners, many of whom were local women themselves, although none of whom participated in the interviews. T The working group affirmed the findings and interpretations were consistent with what they had seen in the region.

## 4 Results

The following section describes the key findings from the exploratory research. Namely, we report on three key areas: the representation of women within farming practice and decision-making, perceptions of recent migrants in the region (intersectional feminism), and finally the perceived role of women during times of drought. These areas were drawn from the codes that were most prevalent in the interviews (Appendix B).

## 4.1 Representation of women

"[I]f [women] want to be seen as equal and if we want to have the same opportunities, then go forth and talk about yourself as a farmer rather than congratulating yourself for being a farmer." (Study Participant 1)

To understand how women think about their representation as farmers, we first needed to examine their beliefs about roles and power within farming communities. Although the image of the iconic Australian farmer has historically been a white man (Rodriguez Castro and Pini, 2022), participants sensed that women around them were more readily adopting the title of 'farmer'

in the past several years than they had previously done, confirming a trend seen in other studies (Shisler and Sbicca, 2019; Sheridan and Newsome, 2021; Rodriguez Castro and Pini, 2022). According to participants, women's adoption of the farmer identity was a slow-moving upward trend rather than a major, sudden one, and there was still a wide representation gap between genders.

      "I don't think in the industry … and the region that I work in, that many [women] do identify themselves as farmers.

It's still a very male dominated industry. So I would like to see changes, but I don't think there has been." (Study Participant 6)

Many participants spoke of an expansion of roles in what it means to be a farmer that they had seen in the past decade. One participant (number 4) suggested that she believed women were more prone to accept the title of 'farmers' because of the credibility associated with that identity. This also carried with it a sense of empowerment and higher perceived legitimacy to

discuss issues of water:

      "Sometimes you do see people describe them as themselves, this farmer's wife, it just depends. … It's a bit of everything in the broad acre [farming community]. I'd say it's more likely that that you'd have women say, No, no, I'm a farmer, because they'll… be counted. They'll be more inclined to stand up and be counted then." (Study Participant 2)

This aspect of being 'counted' refers to having a credible and legitimate voice on issues of farming in the community. This suggests that these women are accepting the mantle of farmer for the utility of the identity rather than for internal identity characteristics alone.

Perceived representation of women within farming was tied to beliefs about underlying power dynamics within different types of farming, particularly differences in dryland farming family roles compared to irrigated agriculture. Irrigators or water users

were seen as having more decision-making power than dryland farmers, despite both being impacted by water availability. This is due to having access to an alternative source of supply (water entitlements) not available to dryland farmers.

      "Often it would be the man who's the decision maker. In the selling [of] water space that would be an irrigator who has it rather than a dryland one." (Study Participant 3)

In contrast to irrigated agriculture, the family farming practices of dryland report women having greater empowerment over

decision-making on matters of the farm and water (e.g. Alston, 2021). Women in dryland communities tend to be well-educated and as some participants noted, more visible in decisions around farming and water. However, common succession practices

in the region means farms and their management often fall to sons following high school (Sheridan et al., 2021; Carolan, 2018). As such, they are trained to take on this career path from early adolescent. Daughters of farming families however are often sent away from the farm to gain a university education in an often non-agricultural career path. It is not unusual to see the daughters within farming families return to their hometowns to start new careers or marry into another farming family, but with a degree in tow (Sheridan et al., 2021). This suggests that women who do return to their farms, or marry into one, often take on management of the financials, customers, partners in the business. This is more commonly seen in dryland farming, and as some participants argue, is a reason that women are able to have some decision-making power in the running of the farming practice.

**4.2 "Us and Them" – Perceptions of Migrant Farmers among Settler Farming Women**

Perhaps the most surprising finding of the interviews (noting that the interviews all involved settler women) was the strength with which migration trends had created a divide between those with farming families in the region for generations compared to those that had recently come into the area. Migration has seemingly ebbed and flowed in the region, with an increase in overseas migration leading up to the 2020 COVID-19 pandemic, dropping away during Australian border closures, and only starting to pick up at time of writing (2023). There have been severe worker shortages within agricultural regions due to the increasing reliance on overseas workers and recent migrants to aid in harvesting of produce. Despite a drop in migrant workers at this time, there were nevertheless first-generation migrant families that had moved into farming practice within the region in the last 20 years. This is as a direct result of the Australian Government's strategy to increase migration to regional areas, offering special visas with the requirement for longer stays. Not only does this create questionable exploitative practices within Agricultural regions (Coates et al., 2023), but it has also subsequently created a distinction between those that have had farming families in the region for generations and those that had recently settled into the area. This distinction was particularly prevalent along the irrigated agriculture versus dryland farming characteristic. Participants mentioned that many migrant families undertook irrigated farming, whereas "Australian" families (as described by participants) were predominantly in dryland. This distinction was noted by one of the interviewees as a *gendered* difference that pointed to traditional gender roles:

> "[T]he men [in irrigated farming with non-English speaking backgrounds] will have a strong belief that it's up to them to provide… for the family" (Study Participant 2)

Some participants put down this as a difference between dryland and irrigated farming when it comes to gendered decision-making. For example, the participant who is a financial advisor relays that in their experience of working with many irrigation families across the region, the decisions on financial matters and any trading of water entitlements was mostly borne by men. Furthermore, there is a strong representation of men on water issues in community meetings with many women feeling disempowered to speak on topics of water, due to their role not encompassing 'farming' within its traditional remit.

Gender norms interact with existing dynamics between cultural background, oftentimes connected to perceived education norms in different communities. Differences in educational norms between farming type has been noted in literature elsewhere (Sheridan et al., 2023), but interviewees in the Mallee also recognised further differences in educational norms for women

between settler farmer families and migrant ones. Many of the irrigated areas that were not corporatized, were seen by participants to be occupied by migrant farming families that carried traditional gender norms within their family units. A participant stated that cultural differences reflected a change in decision-making power as highlighted by the quote below.

"Some European cultures and I'm thinking particularly Greek, Italian, Turkish, the boys are considered more valuable than the girls. … where there's lots of need for labour at particular times of the year, the girls will actually have to leave school earlier to help with picking or help with something, whereas the [boys' value of education] has been more strongly pointed out, perhaps than girls. And again, I'd say that's the opposite to dry land because in dry land, often there's an expectation that the boys in the family will end up being the farmers, so we better get the girls educated." (Study Participant 2)

Although this is a generalised statement that may not reflect all migrant families within farming, it presents an insight into the experience of this participant who works closely with families across the region. Therefore, in reading the discussion on agriculture and dryland farming in the previous section, it cannot be considered in isolation to the migrant dynamic emerging in the area. The migrant dynamic refers to the recent influx of recent migrants settling in the area and entering farming practice. There was a 'push back' from participants in the dryland farms to subscribe to traditional gender norms, using examples such 245 as the joint decision-making at home with non-women partners, or between their men-and-women parents. There are nevertheless distinct differences in the roles that men and women take in dryland farming. In these spaces, women are more likely to be present in meetings, especially those related to financial matters. In irrigated agriculture, some participants highlighted that many women often did not know much about farming or finances, including whether they should sell water entitlements. The desired gender dynamics were realised (or at the very least, perceived to be realised) in their dryland 250 communities, which provided them with a sense of superiority over migrant families.

These include the sense that women could have strong decision-making capacity within farmland practices representing a cultural reformation of the white farming imaginary set in a settler-colonial basis described by Rodriguez Castro and Pini (2022) in their analysis of the Invisible Farmer project. We posit whether the perceived lack of women's voices and legitimacy in migrant farming threatened what settler women wanted to achieve in their communities. As such, although traditional gender 255 norms had been prevalent in farming communities, there is also arguably a 'desired' gender norm permeating recent generations of families. These desired norms relate strongly to a second wave feminism push: particularly in the examination of how traditional femininity is argued to create psychological oppression in women, and in turn, idealise identities that counter femininities in masculine-dominated spaces. In doing so, this creates racialized imaginaries of white farming that continue to uphold colonial practice. These racialised undercurrents are consistent with the ideas of white feminism in the 1970s and 1980s 260 while (as highlighted by Pini et al. 2021) also exhibiting elements of the 'Girl Boss' movement seen in the 2010s (Cavallo and Collins, 2023). Desired gender norms as expressed by participants (and cross-checking with other studies undertaken in the region, eg. Castro and Pini, 2022; Sheridan) include the rejection of traditional feminine roles, particularly in pushing back on the 'cult of domesticity' that defines women as being purely wife and mother. In addition to this, a desired norm of having equitable decision-making between partners, and also sharing of domestic responsibilities. These created 'ideals' of women in

farming can then explain the white feminist discourse that permeates many of the discussions with those in dryland, as coveted positions of leadership in family dynamics are upheld as distinctions of import that separate their communities from 'the other'. This lies in tandem with an ongoing aspect of the increasing racialisation notions of a "local" as being white, despite many generations of migrants and Indigenous people forming an important on-going aspect of community (Stead et al., 2022)..

### 4.3 Perceptions of the role of women in community cohesion during drought

La Niña weather patterns persisted throughout the Mallee region from 2020-2023, which resulted in higher-than-average rainfalls, and even floods (Bureau of Meteorology, nd). Despite this, study participants still held on to memories of the drought closely and these had informed their perception of climate risk. The participants had not 'forgotten' the drought, reinforcing the saliency and impact of these experiences.

> "So, drought might sort of trickle in. It might hit us like a ton of bricks. But in small communities, you see it so
prominently in our agricultural industry because everyone feels it. The water that you use across, … towns is really important as well, and everyone is very much aware and conscious of what they use." (Study Participant 1)

The drought examples that were mentioned by participants referred to direct effects on family health and livelihood. Participants mentioned drought has a 'lag,' where it is difficult to see its effects until time has passed. They associated much of this lag with the caring roles that they had, mostly related to children. A participant describes the harrowing effects of not
having water to properly bathe her children, resulting in skin conditions throughout the family. Another mentions the distressing scenes of witnessing her father needing to sell off the farm due to drought. Participants also mention observing arguments between couples when required financially to sell off water entitlements. Although there were many government schemes and funding available, it was often women who were the ones to organise the paperwork associated with accessing these grants. Once the drought ended, it was noted that these grants subsequently tended to dry up, despite the delayed and
long-term effects of the drought.

One of the themes that emerged from the interviews was the sense of empowerment women felt from their role in the community and accessing drought relief funds to foster community cohesion and resilience during difficult times (Twigg, 2021).

> "Women are often more concerned for the community impacts of drought… so when so when the drought funding
comes and it goes to the community associations… It goes into those… women driven organisations… And it's the women who put the proposals forward on the drought relief events and those sorts of things, they have the interest there to do that." (Study Participant 4)

In addition to this, women's off-farm income has been found to sustain the family during times of drought, especially in dryland farming when women are often educated in off-farm/service work (Alston and Whittenbury, 2013). These are elements
that directly affect women's roles as carers in the family, and the administrative burdens that are associated with relief assistance. Conversely, during the drought periods, although participants felt that women were often relied upon to apply for

grants and to support the family through off-farm work, they also stated that women tended to have a limited say in whether their water licenses were sold.

In considering the previous section on traditional gender roles (and the 'push back' from some in dryland), it is surprising that the caregiving aspects are those mentioned by participants. This is a feature that is also reflected in the literature ie. that it is not unusual for women in dryland farms to take traditional roles in the home such as primary care giver, domestic duties and general emotional support while men take on the role of 'farmer' (Alston, 2021; Shisler and Sbicca, 2019; Stehlik et al., 2000). Similarly, the traditional gender role also encompasses the task that women take on to maintain community social cohesion. As such, the importance of community bonds has not previously been captured as a perceived divider between settler and migrant farmers, particularly related to gender. Participants mentioned that irrigated agriculture did not have as strong a community bond, due to both the changing nature of the demographics in the community (migrant families coming and going) and the increasing corporatisation of farming leading to fewer women creating community cohesion. Although not mentioned by participants, the perception that irrigated agriculture and by extension migrant farmers have less of a community bond could be impacted by timing. Settler communities have generational connections to their and other families in the region that more recently established migrant families do not have access to. Participants believed that looser community bonds could negatively impact disaster resilience. This view is reflected in the literature, as the role of women in establishing and driving these bonds has been shown to be key to rebuilding communities during times of crisis (Lester et al., 2022). In this way, 'caring for the community' was positioned as an asset that dryland (settler) women could bring to farming communities in ways that migrants and men were unlikely to do. Once again, this creates an overarching perception of 'othering,' a social process by which an individual or group's identity is considered lacking and may be subject to discrimination by a dominant in-group (Staszak, 2008; Dervin, 2015). Othering has been examined in the contexts of healthcare, politics, immigration and belongingness (John et al., 2004; Allan Laine Kagedan, 2020; Udah and Singh, 2019). In healthcare contexts, for example, othering can manifest through racializing explanations that affect patient provider interactions and result in differential access to care (Johnson et al., 2004). Similarly, we posit whether this also produces differential access to community support during times of crisis.

**5 Conclusion**

This exploratory study focused on the voice settler women perceive to have on topics of water and drought in farming. In the Mallee region, settler women are experiencing a tenuous but growing connection with farming as an identity which lies in conjunction with their ongoing role in "caring for the community". By adopting 'farmer' status while also taking on roles to foster community cohesion, women are disrupting traditional notions of what it means to be a farmer by performing care-work ,and bolstering their community's resilience against environmental change. With these identity changes come increasing empowerment voice concerns over issues of drought and water rights albeit with the added administrative loads related to applying for grants and bringing in alternative sources of income. In line with previous studies (Carolan, 2018; Sheridan et al., 2021; Alston and Whittenbury, 2013), interviews highlighted an ongoing trend with gendered farming succession which

encouraged women to go to University and develop a career outside of the family farm, while sons were provided with the

tools and training needed to take over the farming business. There are reports that this tendency is beginning to shift, as a result of growing awareness of women's representation and equity in farming, thus challenging gender norms related to succession practices.

The most alarming finding within this exploratory study was the otherness expressed in the interviews towards recent migrant farmers (within irrigated agriculture) from those that have been farming for many generations in dryland. It reflects a dynamic

that is more nuanced than purely along gender lines, but also highlights aspects of who deserves to be a farmer and who performs gender relations 'right'. The issues related to recent migrant and settler colonial farmers is one that has been raised by Barbara Pini and colleagues (2021), who discuss the under-exploration of this topic in the rural sociology literature. Their analysis of publications in the last 20 years had highlighted a burgeoning interest in white women's experiences, with little mention or emphasis on racial inequality and class difference inherent in such environments. Our article acts as a starting point

to begin to address these gaps in the literature, while also being cognisant of not perpetuating colonial settler predominance. Layers of oppression are evident in the study, one layer of women in farming as being 'non-dominant', and the migrant women for the intersecting factors of being women and recent migrants into the area.

This study into the role of women and their identity related to farming should come with a wide range of caveats. To begin with, the small sample size limits the generalizability of the study. It nevertheless provides insights into the exploration of the

research question that can be used to supplement existing literature. Agricultural farming businesses have been predominantly the realm of white settler-colonial peoples and as such, the participants in the study have been overwhelmingly in this category. As much of water decision-making is getting done at the farming level, among corporations, irrigators and white farming families, this carries with it a certain amount of elite status within these discussions compared to Indigenous communities who have their own struggles with retaining water rights. Paying attention to non-dominant voices is key to building resilience in

response to environmental challenges, such as increasing droughts and floods in the region. This does not end at only women, but importantly includes intersectional realms of migrant women, and Indigenous women. Further research is needed with a focus on the intersection between Indigenous groups, migration and gender within these farming communities to provide a more comprehensive view of our drought landscapes.

Ethics approval has been obtained from the University of Melbourne Ethics Committee (ethics no. 2022-23417-25453-4).

**Competing interests**: The contact author has declared that none of the authors has any competing interests.

**Appendix A**

**Interview Questions/Themes**

**Project: Women's voices in the face of drought**

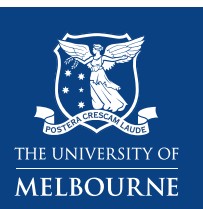

Where: Either in person or online. Dependent on preference of participant/ travel restrictions

How long: 20-30 minutes

Voice recorded for future transcribing. This is outlined in consent form.

This interview is planned as semi-structured, and therefore the questions and themes below could shift with discussion with the participant. However, they nevertheless represent how discussion will be guided. This interview approach takes on a qualitative interviewing method which aims to be flexible, responding to the direction in which interviewees take the interview.

- What voice do women currently have on topics of water and drought? How does this link to the issue of climate
change?
- Are women voices often considered as decision-makers in the area of drought resilience and water management?
- In what ways have women been disempowered in discursive practices in drought discussions?
- How do women frame risks of drought?
- Are there barriers in place for women to have a greater say over water issues? If so, what are these barriers?


**Interview guide**

*Topics: voices of women on drought, decision-making in drought, empowerment in decision-making, risks perceptions of*
*drought, Barriers in place*

*I'll be video recording and then deleting exactly one week after this. So I won't be passing on information from this interview through to Rebecca Wells.*

*How would you describe your main occupation? Do you have any links to farming? Is it dryland or irrigation farming?*

Introduction question: Please tell me about when your interest in drought and water issues began?

Do you feel heard, as a woman, on drought issues? (What sorts of things make you feel heard?)


Do you feel like you can make direct change on issues of water and drought? (What sorts of changes would you implement?)

Water is seen as a contentious topic in this region. Do you see many women around you in decision-making positions when it comes to water? "Farmer's wives"?


How do you think you see drought and water issues differently from those around you?

What barriers do you think may be in place stopping women from having a stronger influence on drought policy and discussions in the region?


Would you like to know the outcomes of this study?

Use of probing questions:
- Could you say some more about that?
- You said earlier that you prefer not to X. Could you say what kinds of things have put you off X?
- What did you do then?
- How did X react to what you said?
- What effect did X have on you?


Appendix B

**Codes used in analysis**

**Structural**

Corporatisation of farming leading to less women in decision-making

Difference between dryland and irrigation

Lack of water

**Social**

Cultural Differences in Women's Roles in Farming

Differences in ways that men and women approach drought

Feeling of disempowerment over drought and water issues

Feeling of empowerment over drought and water issues

Feeling of Empowerment of Property and Land issues

Expectations of Women

Gendered Succession Planning

Lack of masculine interest in the environment

Male dominated decision making

Representation of women

Women's concern about community effects of drought

Women's identity

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
